# EXPRESSIVE POWER OF RECURRENT NEURAL NETWORKS

**Valentin Khrulkov**
Skolkovo Institute of Science and Technology
valentin.khrulkov@skolkovotech.ru

**Alexander Novikov**
National Research University
Higher School of Economics
Institute of Numerical Mathematics RAS
novikov@bayesgroup.ru

**Ivan Oseledets**
Skolkovo Institute of Science and Technology
Institute of Numerical Mathematics RAS
i.oseledets@skoltech.ru

## ABSTRACT

Deep neural networks are surprisingly efficient at solving practical tasks, but the theory behind this phenomenon is only starting to catch up with the practice. Numerous works show that depth is the key to this efficiency. A certain class of deep convolutional networks – namely those that correspond to the Hierarchical Tucker (HT) tensor decomposition – has been proven to have exponentially higher expressive power than shallow networks. I.e. a shallow network of exponential width is required to realize the same score function as computed by the deep architecture. In this paper, we prove the expressive power theorem (an exponential lower bound on the width of the equivalent shallow network) for a class of recurrent neural networks – ones that correspond to the Tensor Train (TT) decomposition. This means that even processing an image patch by patch with an RNN can be exponentially more efficient than a (shallow) convolutional network with one hidden layer. Using theoretical results on the relation between the tensor decompositions we compare expressive powers of the HT- and TT-Networks. We also implement the recurrent TT-Networks and provide numerical evidence of their expressivity.

## 1 INTRODUCTION

Deep neural networks solve many practical problems both in computer vision via Convolutional Neural Networks (CNNs) (LeCun et al. (1995); Szegedy et al. (2015); He et al. (2016)) and in audio and text processing via Recurrent Neural Networks (RNNs) (Graves et al. (2013); Mikolov et al. (2011); Gers et al. (1999)). However, although many works focus on expanding the theoretical explanation of neural networks success (Martens & Medabalimi (2014); Delalleau & Bengio (2011); Cohen et al. (2016)), the full theory is yet to be developed.

One line of work focuses on *expressive power*, i.e. proving that some architectures are more expressive than others. Cohen et al. (2016) showed the connection between Hierarchical Tucker (HT) tensor decomposition and CNNs, and used this connection to prove that deep CNNs are exponentially more expressive than their shallow counterparts. However, no such result exists for Recurrent Neural Networks. The contributions of this paper are three-fold.

1. We show the connection between recurrent neural networks and Tensor Train decomposition (see Sec. 4);

2. We formulate and prove the expressive power theorem for the Tensor Train decomposition (see Sec. 5), which – on the language of RNNs – can be interpreted as follows: to (exactly) emulate a recurrent neural network, a shallow (non-recurrent) architecture of exponentially larger width is required;

3. Combining the obtained and known results, we compare the expressive power of recurrent (TT), convolutional (HT), and shallow (CP) networks with each other (see Table 2).

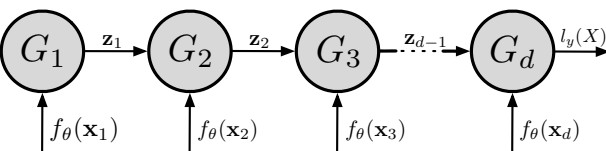

Figure 1: Recurrent-type neural architecture that corresponds to the Tensor Train decomposition. Gray circles are bilinear maps (for details see Section 4).

## 2 DEEP LEARNING AND TENSOR NETWORKS

In this section, we review the known connections between tensor decompositions and deep learning and then show the new connection between Tensor Train decomposition and recurrent neural networks.

Suppose that we have a classification problem and a dataset of pairs $\{(X^{(b)}, y^{(b)})\}_{b=1}^{N}$. Let us assume that each object $X^{(b)}$ is represented as a sequence of vectors

$$X^{(b)} = (\mathbf{x}_1, \mathbf{x}_2, \ldots \mathbf{x}_d), \quad \mathbf{x}_k \in \mathbb{R}^n, \tag{1}$$

which is often the case. To find this kind of representation for images, several approaches are possible. The approach that we follow is to split an image into patches of small size, possibly overlapping, and arrange the vectorized patches in a certain order. An example of this procedure is

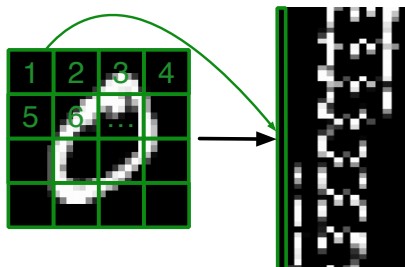

Figure 2: Representation of an image in the form of Eq. (1). A window of size $7 \times 7$ moves across the image of size $28 \times 28$ extracting image patches, which are then vectorized and arranged into a matrix of size $49 \times 16$.

presented on Fig. 2.

We use lower-dimensional *representations* of $\{\mathbf{x}_k\}_{k=1}^{d}$. For this we introduce a collection of parameter dependent feature maps $\{f_{\theta_\ell} : \mathbb{R}^n \to \mathbb{R}\}_{\ell=1}^{m}$, which are organized into a representation map

$$f_\theta : \mathbb{R}^n \to \mathbb{R}^m.$$

A typical choice for such a map is

$$f_\theta(\mathbf{x}) = \sigma(A\mathbf{x} + b),$$

that is an affine map followed by some nonlinear activation $\sigma$. In the image case if $X$ was constructed using the procedure described above, the map $f_\theta$ resembles the traditional convolutional maps – each image patch is projected by an affine map with parameters shared across all the patches, which is followed by a pointwise activation function.

Score functions considered in Cohen et al. (2016) can be written in the form

$$l_y(X) = \langle \mathcal{W}_y, \Phi(X) \rangle, \tag{2}$$

where $\Phi(X)$ is a *feature tensor*, defined as

$$\Phi(X)^{i_1 i_2 \ldots i_d} = f_{\theta_{i_1}}(\mathbf{x}_1) f_{\theta_{i_2}}(\mathbf{x}_2) \ldots f_{\theta_{i_d}}(\mathbf{x}_d), \tag{3}$$

and $\mathcal{W}_y \in \mathbb{R}^{m \times m \times \ldots m}$ is a trainable weight tensor. Inner product in Eq. (2) is just a total sum of the entry-wise product of $\Phi(X)$ and $\mathcal{W}_y$. It is also shown that the hypothesis space of the form Eq. (2) has the universal representation property for $m \to \infty$. Similar score functions were considered in Novikov et al. (2016); Stoudenmire & Schwab (2016).

Storing the full tensor $\mathcal{W}_y$ requires an exponential amount of memory, and to reduce the number of degrees of freedom one can use a *tensor decompositions*. Various decompositions lead to specific network architectures and in this context, expressive power of such a network is effectively measured by *ranks* of the decomposition, which determine the complexity and a total number of degrees of freedom. For the Hierarchical Tucker (HT) decomposition, Cohen et al. (2016) proved the expressive power property, i.e. that for almost any tensor $\mathcal{W}_y$ its HT-rank is exponentially smaller than its CP-rank. We analyze Tensor Train-Networks (TT-Networks), which correspond to a recurrent-type architecture. We prove that these networks also have exponentially larger representation power than shallow networks (which correspond to the CP-decomposition).

## 3 TENSOR FORMATS REMINDER

In this section we briefly review all the necessary definitions. As a $d$-dimensional tensor $\mathcal{X}$ we simply understand a multidimensional array:

$$\mathcal{X} \in \mathbb{R}^{n_1 \times n_2 \times \ldots \times n_d}.$$

To work with tensors it is convenient to use their *matricizations*, which are defined as follows. Let us choose some subset of axes $s = \{i_1, i_2 \ldots i_{m_s}\}$ of $\mathcal{X}$, and denote its compliment by $t = \{j_1, j_2 \ldots j_{d-m_s}\}$, e.g. for a 4 dimensional tensor $s$ could be $\{1, 3\}$ and $t$ is $\{2, 4\}$. Then matricization of $\mathcal{X}$ specified by $(s, t)$ is a matrix

$$\mathcal{X}^{(s,t)} \in \mathbb{R}^{n_{i_1} n_{i_2} \ldots n_{i_{m_s}} \times n_{j_1} n_{j_2} \ldots n_{j_{d-m_s}}},$$

obtained simply by transposing and reshaping the tensor $\mathcal{X}$ into matrix, which in practice e.g. in `Python`, is performed using `numpy.reshape` function. Let us now introduce tensor decompositions we will use later.

### 3.1 CANONICAL

Canonical decomposition, also known as CANDECOMP/PARAFAC or CP-decomposition for short (Harshman (1970); Carroll & Chang (1970)), is defined as follows

$$\mathcal{X}^{i_1 i_2 \ldots i_d} = \sum_{\alpha=1}^{r} \mathbf{v}_{1,\alpha}^{i_1} \mathbf{v}_{2,\alpha}^{i_2} \ldots \mathbf{v}_{d,\alpha}^{i_d}, \quad \mathbf{v}_{i,\alpha} \in \mathbb{R}^{n_i}. \tag{4}$$

The minimal $r$ such that this decomposition exists is called the *canonical* or *CP-rank* of $\mathcal{X}$. We will use the following notation

$$\operatorname{rank}_{CP} \mathcal{X} = r.$$

When $\operatorname{rank}_{CP} \mathcal{X} = 1$ it can be written simply as

$$\mathcal{X}^{i_1 i_2 \ldots i_d} = \mathbf{v}_1^{i_1} \mathbf{v}_2^{i_2} \ldots \mathbf{v}_d^{i_d},$$

which means that modes of $\mathcal{X}$ are perfectly separated from each other. Note that storing all entries of a tensor $\mathcal{X}$ requires $O(n^d)$ memory, while its canonical decomposition takes only $O(dnr)$. However, the problems of determining the exact CP-rank of a tensor and finding its canonical decomposition are NP-hard, and the problem of approximating a tensor by a tensor of lower CP-rank is ill-posed.

### 3.2 TENSOR TRAIN

A tensor $\mathcal{X}$ is said to be represented in the Tensor Train (TT) format (Oseledets (2011)) if each element of $\mathcal{X}$ can be computed as follows

$$\mathcal{X}^{i_1 i_2 \ldots i_d} = \sum_{\alpha_1=1}^{r_1} \sum_{\alpha_2=1}^{r_2} \ldots \sum_{\alpha_{d-1}=1}^{r_{d-1}} G_1^{i_1 \alpha_1} G_2^{\alpha_1 i_2 \alpha_2} \ldots G_d^{\alpha_{d-1} i_d}, \tag{5}$$

where the tensors $G_k \in \mathbb{R}^{r_{k-1} \times n_k \times r_k}$ ($r_0 = r_d = 1$ by definition) are the so-called *TT-cores*. The element-wise minimal ranks $\mathbf{r} = (r_1, \ldots r_{d-1})$ such that decomposition (5) exists are called TT-ranks

$$\mathrm{rank}_{TT} \, \mathcal{X} = \mathbf{r}.$$

Note that for fixed values of $i_1, i_2 \ldots, i_d$, the right-hand side of Eq. (5) is just a product of matrices

$$G_1[1, i_1, :] G_2[:, i_2, :] \ldots G_d[:, i_d, 1].$$

Storing $\mathcal{X}$ in the TT-format requires $O(dnr^2)$ memory and thus also achieves significant compression of the data. Given some tensor $\mathcal{X}$, the algorithm for finding its TT-decomposition is constructive and is based on a sequence of Singular Value Decompositions (SVDs), which makes it more numerically stable than CP-format. We also note that when all the TT-ranks equal to each other

$$\mathrm{rank}_{TT} \, \mathcal{X} = (r, r, \ldots, r),$$

we will sometimes write for simplicity

$$\mathrm{rank}_{TT} \, \mathcal{X} = r.$$

### 3.3 HIERARCHICAL TUCKER

A further generalization of the TT-format leads to the so-called Hierarchical Tucker (HT) format. The definition of the HT-format is a bit technical and requires introducing the *dimension tree* (Grasedyck, 2010, Definition 3.1). In the next section we will provide an informal introduction into the HT-format, and for more details, we refer the reader to Grasedyck (2010); Grasedyck & Hackbusch (2011); Hackbusch (2012).

## 4 ARCHITECTURES BASED ON TENSOR DECOMPOSITIONS

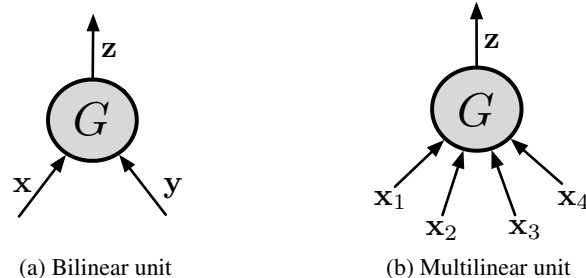

(a) Bilinear unit      (b) Multilinear unit

Figure 3: Nodes performing multilinear map of their inputs. $d$-linear unit is specified by a $d+1$ dimensional core $G$.

To construct the tensorial networks we introduce *bilinear* and *multilinear* units, which perform a bilinear (multilinear) map of their inputs (see Fig. 3 for an illustration). Suppose that $\mathbf{x} \in \mathbb{R}^n, \mathbf{y} \in \mathbb{R}^m$ and $G \in \mathbb{R}^{n \times m \times k}$. Then a bilinear unit $G$ performs a bilinear map $G : \mathbb{R}^n \times \mathbb{R}^m \to \mathbb{R}^k$, defined by the formula

$$G(\mathbf{x}, \mathbf{y}) = \mathbf{z},$$
$$\mathbf{z}^k = \sum_{i,j} G^{ijk} \mathbf{x}^i \mathbf{y}^j. \tag{6}$$

Similarly, for $\mathbf{x}_1 \in \mathbb{R}^{n_1}, \ldots \mathbf{x}_d \in \mathbb{R}^{n_d}$, a multilinear unit $G \in \mathbb{R}^{n_1 \times n_2 \times \ldots \times n_d \times n_j}$ defines a multilinear map $G : \prod_{k=1}^d \mathbb{R}^{n_k} \to \mathbb{R}^{n_j}$ by the formula

$$G(\mathbf{x}_1, \mathbf{x}_2, \ldots, \mathbf{x}_d) = \mathbf{z}$$
$$\mathbf{z}^j = \sum_{i_1, i_2, \ldots, i_d} G^{i_1 i_2 \ldots i_d j} \mathbf{x}_1^{i_1} \mathbf{x}_2^{i_2} \ldots \mathbf{x}_d^{i_d}. \tag{7}$$

In the rest of this section, we describe how to compute the score functions $l_y(X)$ (see Eq. (1)) for each class label $y$, which then could be fed into the loss function (such as cross-entropy). The architecture we propose to implement the score functions is illustrated on Fig. 1. For a vector $\mathbf{r} = (r_1, r_2, \ldots r_{d-1})$ of positive integers (rank hyperparameter) we define bilinear units

$$G_k \in \mathbb{R}^{r_{k-1} \times m \times r_k},$$

with $r_0 = r_d = 1$. Note that because $r_0 = 1$, the first unit $G_1$ is in fact just a linear map, and because $r_d = 1$ the output of the network is just a number. On a step $k \geq 2$ the representation $f_\theta(\mathbf{x}_k)$ and output of the unit $G_{k-1}$ of size $r_k$ are fed into the unit $G_k$. Thus we obtain a recurrent-type neural network with multiplicative connections and without non-linearities.

To draw a connection with the Tensor Train decomposition we make the following observation. For each of the class labels $y$ let us construct the tensor $\mathcal{W}_y$ using the definition of TT-decomposition (Eq. (5)) and taking $\{G_k\}_{k=1}^d$ used for constructing $l_y(X)$ as its TT-cores. Using the definition of the Eq. (3) we find that the score functions computed by the network from Fig. 1 are given by the formula

$$l_y(X) = \sum_{i_1, i_2, \ldots i_d} W_y^{i_1 i_2 \ldots i_d} \Phi(X)^{i_1 i_2 \ldots i_d}, \tag{8}$$

which is verified using Eq. (5) and Eq. (3). Thus, we can conclude that the network presented on Fig. 1 realizes the TT-decomposition of the weight tensor. We also note that the size of the output of the bilinear unit $G_k$ in the TT-Network is equal to $r_k$, which means that the TT-ranks correspond to the *width* of the network.

Let us now consider other tensor decompositions of the weight tensors $\mathcal{W}_y$, construct corresponding network architectures, and compare their properties with the original TT-Network.

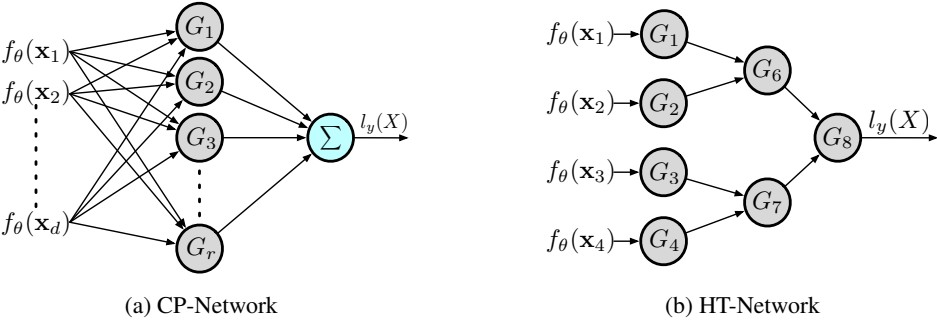

(a) CP-Network  (b) HT-Network

Figure 4: Examples of networks corresponding to various tensor decompositions.

A network corresponding to the CP-decomposition is visualized on Fig. 4a. Each multilinear unit $G_\alpha$ is given by a summand in the formula Eq. (4), namely

$$G_\alpha^{i_1 i_2 \ldots i_d} = \mathbf{v}_{1,\alpha}^{i_1} \mathbf{v}_{2,\alpha}^{i_2} \ldots \mathbf{v}_{d,\alpha}^{i_d}, \quad \alpha \in \{1, \ldots r\}.$$

Note that the output of each $G_\alpha$ in this case is just a number, and in total there are $\mathrm{rank}_{CP} \mathcal{W}_y$ multilinear units. Their outputs are then summed up by the $\Sigma$ node. As before rank of the decomposition corresponds to the width of the network. However, in this case the network is *shallow*, meaning that there is only one hidden layer.

On the Fig. 4b a network of other kind is presented. Tensor decomposition which underlies it is the Hierarchical Tucker decomposition, and hence we call it the HT-Network. It is constructed using a binary tree, where each node other than leaf corresponds to a bilinear unit, and leaves correspond to linear units. Inputs are fed into leaves, and this data is passed along the tree to the root, which outputs a number. Ranks, in this case, are just the sizes of the outputs of the intermediate units. We will denote them by $\mathrm{rank}_{HT} \mathcal{X}$. These are networks considered in Cohen et al. (2016), where the expressive power of such networks was analyzed and was argued that they resemble traditional CNNs. In general Hierarchical Tucker decomposition may be constructed using an arbitrary tree, but not much theory is known in general case.

Our main theoretical results are related to a comparison of the expressive power of these kinds of networks. Namely, the question that we ask is as follows. Suppose that we are given a TT-Network. How complex would be a CP- or HT-Network realizing the same score function? A natural measure of complexity, in this case, would be the rank of the corresponding tensor decomposition. To make transitioning between tensor decompositions and deep learning vocabulary easier, we introduce the following table.

Table 1: Correspondence between languages of Tensor Analysis and Deep Learning.

| Tensor Decompositions | Deep Learning |
|---|---|
| CP-decomposition | shallow network |
| TT-decomposition | RNN |
| HT-decomposition | CNN |
| rank of the decomposition | width of the network |

## 5 THEORETICAL ANALYSIS

In this section we prove the expressive power theorem for the Tensor Train decomposition, that is we prove that given a random $d$-dimensional tensor in the TT format with ranks $\mathbf{r}$ and modes $n$, with probability 1 this tensor will have exponentially large CP-rank. Note that the reverse result can not hold true since TT-ranks can not be larger than CP-ranks: $\operatorname{rank}_{TT} \mathcal{X} \leq \operatorname{rank}_{CP} \mathcal{X}$.

It is known that the problem of determining the exact CP-rank of a tensor is NP-hard.

To bound CP-rank of a tensor the following lemma is useful.

**Lemma 1.** Let $\mathcal{X}^{i_1 i_2 \ldots i_d}$ and $\operatorname{rank}_{CP} \mathcal{X} = r$. Then for any matricization $\mathcal{X}^{(s,t)}$ we have $\operatorname{rank} \mathcal{X}^{(s,t)} \leq r$, where the ordinary matrix rank is assumed.

*Proof.* Proof is based on the following observation. Let

$$\mathcal{A}^{i_1 i_2 \ldots i_d} = \mathbf{v}_1^{i_1} \mathbf{v}_2^{i_2} \ldots \mathbf{v}_d^{i_d},$$

be a CP-rank 1 tensor. Note for any $s, t$

$$\operatorname{rank} \mathcal{A}^{(s,t)} = 1,$$

because $\mathcal{A}^{(s,t)}$ can be written as $\mathbf{u}\mathbf{w}^T$ for some $\mathbf{u}$ and $\mathbf{w}$. Then the statement of the lemma follows from the facts that matricization is a linear operation, and that for matrices

$$\operatorname{rank}(A + B) \leq \operatorname{rank} A + \operatorname{rank} B.$$

$\square$

We use this lemma to provide a lower bound on the CP-rank in the theorem formulated below. For example, suppose that we found some matricization of a tensor $\mathcal{X}$ which has matrix rank $r$. Then, by using the lemma we can estimate that $\operatorname{rank}_{CP} \mathcal{X} \geq r$.

Let us denote $\mathbf{n} = (n_1, n_2 \ldots n_d)$. Set of all tensors $\mathcal{X}$ with mode sizes $\mathbf{n}$ representable in TT-format with

$$\operatorname{rank}_{TT} \mathcal{X} \leq \mathbf{r},$$

for some vector of positive integers $\mathbf{r}$ (inequality is understood entry-wise) forms an *irreducible algebraic variety* (Shafarevich & Hirsch (1994)), which we denote by $\mathcal{M}_\mathbf{r}$. This means that $\mathcal{M}_\mathbf{r}$ is defined by a set of polynomial equations in $\mathbb{R}^{n_1 \times n_2 \ldots n_d}$, and that it can not be written as a union (not necessarily disjoint) of two proper non-empty algebraic subsets. An example where the latter property does not hold would be the union of axes $x = 0$ and $y = 0$ in $\mathbb{R}^2$, which is an algebraic set defined by the equation $xy = 0$. The main fact that we use about irreducible algebraic varieties is that any *proper* algebraic subset of them necessarily has measure 0 (Ilyashenko & Yakovenko (2008)).

For simplicity let us assume that number of modes $d$ is even, that all mode sizes are equal to $n$, and we consider $\mathcal{M}_{\mathbf{r}}$ with $\mathbf{r} = (r, r \ldots r)$, so for any $\mathcal{X} \in \mathcal{M}_{\mathbf{r}}$ we have

$$\mathrm{rank}_{TT} \, \mathcal{X} \leq (r, r, \ldots, r),$$

entry-wise.

As the main result we prove the following theorem

**Theorem 1.** Suppose that $d = 2k$ is even. Define the following set

$$B = \{\mathcal{X} \in \mathcal{M}_{\mathbf{r}} : \mathrm{rank}_{CP} \, \mathcal{X} < q^{\frac{d}{2}}\},$$

where $q = \min\{n, r\}$.

Then

$$\mu(B) = 0,$$

where $\mu$ is the standard Lebesgue measure on $\mathcal{M}_{\mathbf{r}}$.

*Proof.* Our proof is based on applying Lemma 1 to a particular matricization of $\mathcal{X}$. Namely, we would like to show that for $s = \{1, 3, \ldots d - 1\}$, $t = \{2, 4, \ldots d\}$ the following set

$$B^{(s,t)} = \{\mathcal{X} \in \mathcal{M}_{\mathbf{r}} : \mathrm{rank} \, \mathcal{X}^{(s,t)} \leq q^{\frac{d}{2}} - 1\},$$

has measure 0. Indeed, by Lemma 1 we have

$$B \subset B^{(s,t)},$$

so if $\mu(B^{(s,t)}) = 0$ then $\mu(B) = 0$ as well. Note that $B^{(s,t)}$ is an algebraic subset of $\mathcal{M}_{\mathbf{r}}$ given by the conditions that the determinants of all $q^{\frac{d}{2}} \times q^{\frac{d}{2}}$ submatrices of $\mathcal{X}^{(s,t)}$ are equal to 0. Thus to show that $\mu(B^{(s,t)}) = 0$ we need to find at least one $\mathcal{X}$ such that $\mathrm{rank} \, \mathcal{X}^{(s,t)} \geq q^{\frac{d}{2}}$. This follows from the fact that because $B^{(s,t)}$ is an algebraic subset of the irreducible algebraic variety $\mathcal{M}_{\mathbf{r}}$, it is either equal to $\mathcal{M}_{\mathbf{r}}$ or has measure 0, as was explained before.

One way to construct such tensor is as follows. Let us define the following tensors:

$$
\begin{aligned}
G_1^{i_1 \alpha_1} &= \delta_{i_1 \alpha_1}, \quad G_1 \in \mathbb{R}^{1 \times n \times r} \\
G_k^{\alpha_{k-1} i_k \alpha_k} &= \delta_{i_k \alpha_{k-1}}, \quad G_k \in \mathbb{R}^{r \times n \times 1}, k = 2, 4, 6, \ldots, d - 2 \\
G_k^{\alpha_{k-1} i_k \alpha_k} &= \delta_{i_k \alpha_k}, \quad G_k \in \mathbb{R}^{1 \times n \times r}, k = 3, 5, 7, \ldots, d - 1 \\
G_d^{\alpha_{d-1} i_d} &= \delta_{i_d \alpha_{d-1}}, \quad G_d \in \mathbb{R}^{r \times n \times 1}
\end{aligned}
\tag{9}
$$

where $\delta_{i\alpha}$ is the Kronecker delta symbol:

$$
\delta_{i\alpha} = \begin{cases} 1, & \text{if } i = \alpha, \\ 0, & \text{if } i \neq \alpha. \end{cases}
$$

The TT-ranks of the tensor $\mathcal{X}$ defined by the TT-cores (9) are equal to $\mathrm{rank}_{TT} \, \mathcal{X} = (r, 1, r, \ldots, r, 1, r)$.

Lets consider the following matricization of the tensor $\mathcal{X}$

$$\mathcal{X}^{(i_1, i_3, \ldots, i_{d-1}), (i_2, i_4, \ldots, i_d)}$$

The following identity holds true for any values of indices such that $i_k = 1, \ldots, q, \ k = 1, \ldots, d$.

$$
\mathcal{X}^{(i_1, i_3, \ldots, i_{d-1}), (i_2, i_4, \ldots, i_d)} = \sum_{\alpha_1, \ldots, \alpha_{d-1}} G_1^{i_1 \alpha_1} \ldots G_d^{\alpha_{d-1} i_d} =
$$

$$
\sum_{\alpha_1, \ldots, \alpha_{d-1}} \delta_{i_1 \alpha_1} \delta_{i_2 \alpha_1} \delta_{i_3 \alpha_3} \ldots \delta_{i_d, \alpha_{d-1}} = \delta_{i_1 i_2} \delta_{i_3 i_4} \ldots \delta_{i_{d-1} i_d}
\tag{10}
$$

The last equality holds because $\sum_{\alpha_k=1}^{r} \delta_{i_k \alpha_k} \delta_{i_{k+1} \alpha_k} = \delta_{i_k i_{k+1}}$ for any $i_k = 1, \ldots, q$. We obtain that

$$\mathcal{X}^{(i_1, i_3, \ldots, i_{d-1}), (i_2, i_4, \ldots, i_d)} = \delta_{i_1 i_2} \delta_{i_3 i_4} \ldots \delta_{i_{d-1} i_d} = I^{(i_1, i_3, \ldots, i_{d-1}), (i_2, i_4, \ldots, i_d)}, \tag{11}$$

where $I$ is the identity matrix of size $q^{d/2} \times q^{d/2}$ where $q = \min\{n, r\}$.

To summarize, we found an example of a tensor $\mathcal{X}$ such that $\mathrm{rank}_{TT}\, \mathcal{X} \leq \mathbf{r}$ and the matricization $\mathcal{X}^{(i_1, i_3, \ldots, i_{d-1}),(i_2, i_4, \ldots, i_d)}$ has a submatrix being equal to the identity matrix of size $q^{d/2} \times q^{d/2}$, and hence $\mathrm{rank}\, \mathcal{X}^{(i_1, i_3, \ldots, i_{d-1}),(i_2, i_4, \ldots, i_d)} \geq q^{d/2}$.

This means that the canonical $\mathrm{rank}_{CP}\, \mathcal{X} \geq q^{d/2}$ which concludes the proof. $\qquad\square$

In other words, we have proved that for all TT-Networks besides negligible set, the equivalent CP-Network will have exponentially large width. To compare the expressive powers of the HT- and TT-Networks we use the following theorem (Grasedyck, 2010, Section 5.3.2).

**Theorem 2.** For any tensor $\mathcal{X}$ the following estimates hold.

- If $\mathrm{rank}_{TT}\, \mathcal{X} \leq r$, then $\mathrm{rank}_{HT}\, \mathcal{X} \leq r^2$.

- If $\mathrm{rank}_{HT}\, \mathcal{X} \leq r$, then $\mathrm{rank}_{TT}\, \mathcal{X} \leq r^{\log_2(d)/2}$.

It is also known that this bounds are *sharp* (see Buczyńska et al. (2015)). Thus, we can summarize all the results in the following Table 2.

Table 2: Comparison of the expressive power of various networks. Given a network of width $r$, specified in a column, rows correspond to the upper bound on the width of the equivalent network of other type (we assume that the number of feature maps $m$ is greater than the width of the network $r$).

|  | **TT-Network** | **HT-Network** | **CP-Network** |
|---|---|---|---|
| TT-Network | $r$ | $r^{\log_2(d)/2}$ | $r$ |
| HT-Network | $r^2$ | $r$ | $r$ |
| CP-Network | $\geq r^{\frac{d}{2}}$ | $\geq r^{\frac{d}{2}}$ | $r$ |

**Example that requires exponential width in a shallow network**   A particular example used to prove Theorem 1 is not important per se since the Theorem states that TT is exponentially more expressive than CP for almost any tensor (for a set of tensors of measure one). However, to illustrate how the Theorem translates into neural networks consider the following example.

Consider the task of getting $d$ input vectors with $n$ elements each and aiming to compute the following measure of similarity between $\mathbf{x}_1, \ldots, \mathbf{x}_{d/2}$ and $\mathbf{x}_{d/2+1}, \ldots, \mathbf{x}_d$:

$$l(X) = (\mathbf{x}_1^\mathsf{T} \mathbf{x}_{d/2+1}) \ldots (\mathbf{x}_{d/2}^\mathsf{T} \mathbf{x}_d) \tag{12}$$

We argue that it can be done with a TT-Network of width $n$ by using the TT-tensor $\mathcal{X}$ defined in the proof of Theorem 1 and feeding the input vectors in the following order: $\mathbf{x}_1, \mathbf{x}_{d/2+1}, \ldots \mathbf{x}_{d/2}, \mathbf{x}_d$. The CP-network representing the same function will have $n^{d/2}$ terms (and hence $n^{d/2}$ width) and will correspond to expanding brackets in the expression (12).

**The case of equal TT-cores**   In analogy to the traditional RNNs we can consider a special class of Tensor Trains with the property that all the intermediate TT-cores are equal to each other: $G_2 = G_3 = \cdots = G_{d-1}$, which allows for processing sequences of varied length. We hypothesize that for this class exactly the same result as in Theorem 1 holds i.e. if we denote the variety of Tensor Trains with equal TT-cores by $\mathcal{M}_\mathbf{r}^{eq}$, we believe that the following hypothesis holds true:

**Hypothesis 1.** Theorem 1 is also valid if $\mathcal{M}_\mathbf{r}$ is replaced by $\mathcal{M}_\mathbf{r}^{eq}$.

To prove it we can follow the same route as in the proof of Theorem 1. While we leave finding an analytical example of a tensor with the desired property of rank maximality to a future work, we have verified numerically that randomly generated tensors $\mathcal{X}$ from $\mathcal{M}_\mathbf{r}^{eq}$ with $d = 6$, $n$ ranging from 2 to 10 and $r$ ranging from 2 to 20 (we have checked 1000 examples for each possible combination) indeed satisfy $\mathrm{rank}_{CP}\, \mathcal{X} \geq q^{\frac{d}{2}}$.

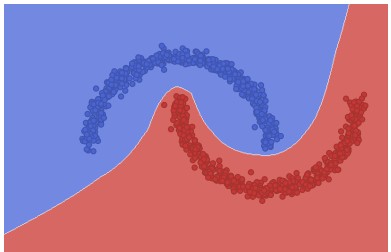 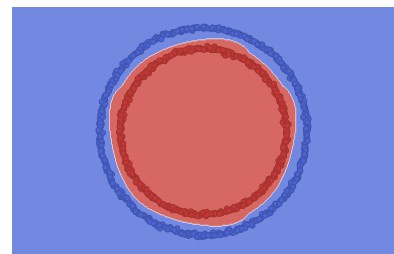

Figure 5: Decision boundaries of the TT-Network on toy 2-D datasets.

## 6 EXPERIMENTS

In this section, we experimentally check if indeed – as suggested by Theorem 1 – the CP-Networks require exponentially larger width compared to the TT-Networks to fit a dataset to the same level of accuracy. This is not clear from the theorem since for natural data, functions that fit this data may lay in the neglectable set where the ranks of the TT- and CP-networks are related via a polynomial function (in contrast to the exponential relationship for all function outside the neglectable set). Other possible reasons why the theory may be disconnected with practice are optimization issues (although a certain low-rank tensor exists, we may fail to find it with SGD) and the existence of the feature maps, which were not taken into account in the theory.

To train the TT- and CP-Networks, we implemented them in `TensorFlow` (Abadi et al. (2015)) and used Adam optimizer with batch size 32 and learning rate sweeping across {4e-3, 2e-3, 1e-3, 5e-4} values. Since we are focused on assessing the expressivity of the format (in contrast to its sensitivity to hyperparameters), we always choose the best performing run according to the training loss.

For the first experiment, we generate two-dimensional datasets with `Scikit-learn` tools 'moons' and 'circles' (Pedregosa et al. (2011)) and for each training example feed the two features as two patches into the TT-Network (see Fig. 5). This example shows that the TT-Networks can implement nontrivial decision boundaries.

For the next experiments, we use computer vision datasets MNIST (LeCun et al. (1990)) and CIFAR-10 (Krizhevsky & Hinton (2009)). MNIST is a collection of 70000 handwritten digits, CIFAR-10 is a dataset of 60000 natural images which are to be classified into 10 classes such as bird or cat. We feed raw pixel data into the TT- and CP-Networks (which extract patches and apply a trainable feature map to them, see Section 2). In our experiments we choose patch size to be $8 \times 8$, feature maps to be affine maps followed by the ReLU activation and we set number of such feature maps to 4. For MNIST, both TT- and CP-Networks show reasonable performance (1.0 train accuracy, 0.95 test accuracy without regularizers, and 0.98 test accuracy with dropout 0.8 applied to each patch) even with ranks less than 5, which may indicate that the dataset is too simple to draw any conclusion, but serves as a sanity check.

We report the training accuracy for CIFAR-10 on Fig. 6. Note that we did not use regularizers of any sort for this experiment since we wanted to compare expressive power of networks (the best test accuracy we achieved this way on CIFAR-10 is 0.45 for the TT-Network and 0.2 for the CP-Network). On practice, the expressive power of the TT-Network is only polynomially better than that of the CP-network (Fig. 6), probably because of the reasons discussed above.

## 7 RELATED WORK

A large body of work is devoted to analyzing the theoretical properties of neural networks (Cybenko (1989); Hornik et al. (1989); Shwartz-Ziv & Tishby (2017)). Recent studies focus on depth efficiency (Raghu et al. (2017); Montufar et al. (2014); Eldan & Shamir (2016); Sutskever et al. (2013)), in most cases providing worst-case guaranties such as bounds between deep and shallow networks width. Two works are especially relevant since they analyze depth efficiency from the viewpoint of tensor decompositions: expressive power of the Hierarchical Tucker decomposition (Cohen et al. (2016)) and its generalization to handle activation functions such as ReLU (Cohen

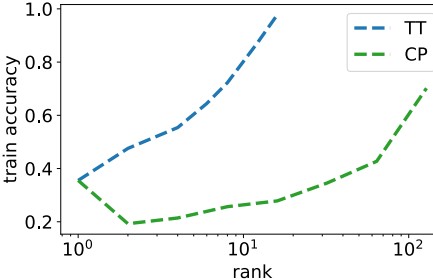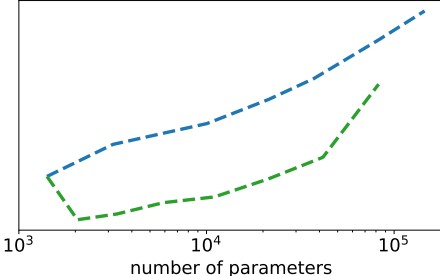

Figure 6: Train accuracy on CIFAR-10 for the TT- and CP-Networks wrt rank of the decomposition and total number of parameters (feature size 4 was used). Note that with rank increase the CP-Networks sometimes perform worse due to optimization issues.

& Shashua (2016)). However, all of the works above focus on feedforward networks, while we tackle recurrent architectures. The only other work that tackles expressivity of RNNs is the concurrent work that applies the TT-decomposition to explicitly modeling high-order interactions of the previous hidden states and analyses the expressive power of the resulting architecture (Yu et al., 2017). This work, although very related to ours, analyses a different class of recurrent models.

Models similar to the TT-Network were proposed in the literature but were considered from the practical point of view in contrast to the theoretical analyses provided in this paper. Novikov et al. (2016); Stoudenmire & Schwab (2016) proposed a model that implements Eq. (2), but with a pre-defined (not learnable) feature map $\Phi$. Wu et al. (2016) explored recurrent neural networks with multiplicative connections, which can be interpreted as the TT-Networks with bilinear maps that are shared $G_k = G$ and have low-rank structure imposed on them.

## 8 CONCLUSION

In this paper, we explored the connection between recurrent neural networks and Tensor Train decomposition and used it to prove the expressive power theorem, which states that a shallow network of exponentially large width is required to mimic a recurrent neural network. The downsides of this approach is that it provides worst-case analysis and do not take optimization issues into account. In the future work, we would like to address the optimization issues by exploiting the Riemannian geometry properties of the set of TT-tensors of fixed rank and extend the analysis to networks with non-linearity functions inside the recurrent connections (as was done for CNNs in Cohen & Shashua (2016)).

## ACKNOWLEDGEMENTS

This study was supported by the Ministry of Education and Science of the Russian Federation (grant 14.756.31.0001).

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
