# OpenReview forum: "Expressive power of recurrent neural networks"
_ICLR.cc/2018/Conference — Accept (Poster)_

### Official Review · AnonReviewer2 · 2017-11-25
**An interesting theoretical paper**

**Rating:** 6
**Confidence:** 5

**Review:**

In this paper, the expressive power of neural networks characterized by tensor train (TT) decomposition, a chain-type tensor decomposition, is investigated. Here, the expressive power refers to the rank of tensor decomposition, i.e., the number of latent components. The authors compare the complexity of TT-type networks with networks structured by CP decomposition, which corresponds to shallow networks. It is proved that the space of TT-type networks with rank O(r)  can be complex as the same as the space of CP-type networks with rank poly(r).

The paper is clearly written and easy to follow.

The contribution is clear and it is distinguished from previous studies.

Though I enjoyed reading this paper, I have several concerns.

1. The authors compare the complexity of TT representation with CP representation (and HT representation). However, CP representation does not have universality (i.e., some tensors cannot be expressed by CP representation with finite rank, see [1]), this comparison may not make sense. It seems the comparison with Tucker-type representation makes much more sense because it has universality.

2. Connecting RNN and TT representation is a bit confusing. Specifically, I found two gaps.
   (a) RNNs reuse the same parameter against all the input x_1 to x_d. This means that G_1 to G_d in Figure 1 are all the same. That's why RNNs can handle size-varying sequences.
   (b) Standard RNNs do not use the multilinear units shown in Figure 3, but use a simple addition of an input and the output from the previous layer (i.e., h_t = f(Wx_t + Vh_{t-1}), where h_t is the t-th hidden unit, x_t is the t-th input, W and V are weights, and f is an activation function.)
Due to the gaps, the analysis used in this paper seems not applicable to RNNs. If this is true, the story of this paper is somewhat misleading. Or, is your theory still applicable?

[1] Hackbusch, Wolfgang. Tensor spaces and numerical tensor calculus. Vol. 42. Springer Science & Business Media, 2012.

---

> ### Author Response · Authors · 2018-01-02
> **Author's response**
>
> >The authors compare the complexity of TT representation with CP representation (and HT representation). However, CP representation does not have universality (i.e., some tensors cannot be expressed by CP representation with finite rank, see [1]), this comparison may not make sense.
>
> We believe that any tensor admits finite CP-rank which for a tensor A of dimension d and mode size n is bounded by n^d. This worst case scenario is obtained by writing A = \sum_{i1 i2 .. id } A_{i1 i2 .. id }e_{i1) \otimes e_{i2) … \otimes e_{id}, that is we write A as a sum of elementary tensors (tensor product basis).
>
> >NNs reuse the same parameter against all the input x_1 to x_d. This means that G_1 to G_d in Figure 1 are all the same. That's why RNNs can handle size-varying sequences.
>
> Thank you for raising this point. We believe that this statement will also hold true and verified it numerically -- in all the experiments with randomly generated tensor in TT format with shared parameters the same permutation as in the proof of Theorem 1 gave us a matrix of maximal rank. We have added a small discussion on this issue to the paper and provided details of the numerical experiment. https://ibb.co/ic0T4w
>
> >Standard RNNs do not use the multilinear units shown in Figure 3, but use a simple addition of an input and the output from the previous layer (i.e., h_t = f(Wx_t + Vh_{t-1}), where h_t is the t-th hidden unit, x_t is the t-th input, W and V are weights, and f is an activation function.)
> Due to the gaps, the analysis used in this paper seems not applicable to RNNs. If this is true, the story of this paper is somewhat misleading. Or, is your theory still applicable?
>
> As we noted in the related work section, [Wu et al, 2016] recently explored RNNs with multiplicative interactions and found them to be quite effective. We can interpret TT-network as a multiplicative RNN from [Wu et a.l, 2016] with two differences: 1) we don’t use an activation function for the recurrent connection 2) we use a general 3-dimensional map defined by a TT-core tensor, while the map in [Wu et al., 2016] can be interpreted as a low-rank approximation of what we used.
> As for the activation function, we think that even without it multiplicative RNNs can be flexible enough to be used in practice and thus their analysis can shed light on the behavior of RNNs in general. Also note, that although the recurrent connection doesn’t have an activation function, the feature map Ф can be arbitrarily complex.
> We also believe that ReLU can be added to the analysis eventually (following the steps of [Cohen et al., 2016] who proved the exponential expressive power of HT-format and then followed up [Cohen and Shashua, 2016] with a generalization of the proof for the networks with activation function), and leave it as a future work.

---

### Official Review · AnonReviewer1 · 2017-11-25
**Important result, but some room for improvement.**

**Rating:** 6
**Confidence:** 3

**Review:**

This paper investigates an expressive power of the tensor train decomposition relative to the CP-decomposition. The result of this paper is interesting and also important from a viewpoint on analysis for the tensor train decomposition.

However, I think there is some room for improvement on this paper. Comments are as follow.

C1.
Could you describe more details about the importance of an irreducible algebraic variety? Especially, it will be nice if authors provide practical examples of tensors in $\mathcal{M}_r$ and tensors not in $\mathcal{M}_r$. The present description about $\mathcal{M}_r$ is too simple and thus I cannot judge whether the restriction on $\mathcal{M}_r$ is critical or not.

C2.
I wonder that the experiment for comparing TT-decomposition and CP-decomposition is fair, since CP-decomposition does not have the universal approximation property. Is it possible to conduct numerical experiments for comparing the ranks directly? For example, given a tensor with known CP-rank, could you measure the TT-rank of the tensor? Such experiments will improve persuasiveness of the main result presented in this paper.

---

> ### Author Response · Authors · 2018-01-02
> **Author's response**
>
> > Could you describe more details about the importance of an irreducible algebraic variety? Especially, it will be nice if authors provide practical examples of tensors in $\mathcal{M}_r$ and tensors not in $\mathcal{M}_r$. The present description about $\mathcal{M}_r$ is too simple and thus I cannot judge whether the restriction on $\mathcal{M}_r$ is critical or not.
>
> Thank you for raising this point. The question of which tensors admit low-rank decompositions is very interesting and nontrivial. Typically tensors are obtained as the values of a function sampled on some uniform grid. For many functions such as polynomials, sin, exp there exist theoretical bounds on the magnitude of the TT-ranks of the resulting tensor, showing that they are small, and if one constructs a linear combinations of such functions we can estimate that TT-ranks (A + B) <= TT-ranks(A) + TT-ranks(B).
> In general, when we sample a smooth function, the smoother function is the lower TT-ranks will be. Moreover if one introduces some small rounding parameter eps, for many tensors in practice it is possible to find a TT decomposition with the relative accuracy eps, but with much smaller ranks. White noise, on the other hand, will have the maximal TT-rank (with probability 1) because of the lack of smoothness or structure.
> This can be thought as an analogy to Fourier series, where to approximate a smooth function with some accuracy only small amount of summands is required. In many applications, TT-ranks are modest and allow for computations with tensors which would be impossible to store explicitly (e.g. they might have 10^30 entries in full format).
>
>
> >I wonder that the experiment for comparing TT-decomposition and CP-decomposition is fair, since CP-decomposition does not have the universal approximation property. Is it possible to conduct numerical experiments for comparing the ranks directly? For example, given a tensor with known CP-rank, could you measure the TT-rank of the tensor? Such experiments will improve persuasiveness of the main result presented in this paper.
>
> Thank you for this suggestion. First of all we would like to note that an arbitrary d-dimensional tensor A with mode size n admits canonical decomposition in the worst case of the rank n^d, which can be obtained in the form
> \sum_{i1 i2 .. id } A_{i1 i2 .. id }e_{i1) \otimes e_{i2) … \otimes e_{id}, that is we just write A in the tensor product basis, which implies that CP-format also has universal approximation property (however CP-rank n^d is clearly impractical).
> As for a comparison between CP-ranks and TT-ranks it can be noted that if CP-rank = R then TT-ranks are bounded by R. This can be explained by the fact that if CP-rank = R then rank of any matricization of the tensor is <= R, and TT-ranks are equal to matrix ranks of particular matricizations. We briefly state it in the beginning of Section 5 and in Table 2.
> Tensors we work with in this paper are too large to be formed explicitly and estimate their CP-rank (although their TT-ranks are small). For small tensors e.g. of size 3 x 3 x 3 x 3 with given TT-ranks we have performed numerous experiments estimating their CP-rank and all the cases we got that they have maximal rank (as claimed in the paper). If you think that this analysis is necessary we will extend Section 6 with the details of this experiment.

---

### Official Review · AnonReviewer3 · 2017-11-29
**Some gap between theory and practice**

**Rating:** 6
**Confidence:** 4

**Review:**

The authors of this paper first present a class of networks inspired by various tensor decomposition models. Then they focus on one particular decompostion known as the tensor train decomposition and points out an analogy between tensor train networks and recurrent neural networks. Finally the authors show that almost all tensor train networks (exluding a set of measure zero) require exponentially large width to represent in CP networks, which is analogous to shallow networks.

While I enjoyed reading the gentle introduction, nice overview of past work, and the theoretical analysis that relates the rank of tensor train networks to that of CP netowkrs, I wasn't sure how to translate the finding into the corresponding neural network models, namely, recurrent neural networks and shallow MLPs.

For example,
 * How does the "bad" example (low TT-rank but exponentially large CP-rank) translate into a recurrent neural network?
 * For both TT-networks and CP-networks, there are multilinear interaction of the inputs/previous hidden states. How precise is the analogy? Can we somehow restrict the interactions to additive ones so that we can exactly recover MLPs or RNNs?

I also did not find the experiments illuminating. First of all the authors need to provide more details about how CP or TT networks are applies to MNIST and CIFAR-10 datasets. For example, the number of input patches and the number of hidden units, etc. In addition, I would like to see the performance of RNNs and MLPs with the same number of units/rank in order to validate the analogy between these networks. Finally I think it makes sense to try some sequence datasets for which RNNs are typically used.

Minor comments:
 * In p7 it would help readers to point out that B^{(s,t)} is an algebraic subset because it is an intersection of M_r and the set of matrices of rank at most q^{d/2} - 1, which is known to be algebraic.

---

> ### Author Response · Authors · 2018-01-02
> **Author's response**
>
> > How does the "bad" example (low TT-rank but exponentially large CP-rank) translate into a recurrent neural network?
> Thank you for this question, we added the interpretation from the neural network point of view into the updated paper. https://ibb.co/eAeZeb
> But note that the particular example is not that important since we proved that the statement of the theorem holds for almost all tensors (i.e. for a set of tensors of measure 1).
>
> > For both TT-networks and CP-networks, there are multilinear interaction of the inputs/previous hidden states. How precise is the analogy? Can we somehow restrict the interactions to additive ones so that we can exactly recover MLPs or RNNs?
> As we noted in the related work section, [Wu et al, 2016] recently explored RNNs with multiplicative interactions and found them to be quite effective. We can interpret TT-network as a multiplicative RNN from [Wu et a.l, 2016] with two differences: 1) we don’t use an activation function for the recurrent connection 2) we use a general 3-dimensional map defined by a TT-core tensor, while the map in [Wu et al., 2016] can be interpreted as a low-rank approximation of what we used.
> As for the activation function, we think that even without it multiplicative RNNs can be flexible enough to be used in practice and thus their analysis can shed light on the behavior of RNNs in general. Also note, that although the recurrent connection doesn’t have an activation function, the feature map Ф can be arbitrarily complex.
> We also believe that ReLU can be added to the analysis eventually (following the steps of [Cohen et al., 2016] who proved the exponential expressive power of HT-format and then followed up [Cohen and Shashua, 2016] with a generalization of the proof for the networks with activation function), and leave it as a future work.
>
> If you think this clarification is important for the understanding, we will extend the paragraph in the related work section dedicated to this issue.
>
> > I also did not find the experiments illuminating. First of all the authors need to provide more details about how CP or TT networks are applies to MNIST and CIFAR-10 datasets. For example, the number of input patches and the number of hidden units, etc.
>
> In our experiments we chose patch size to be 8 x 8, feature maps to be affine maps followed by the ReLU activation and we set the number of such feature maps to 4. We have added this information to the experiments section.
>
> > In addition, I would like to see the performance of RNNs and MLPs with the same number of units/rank in order to validate the analogy between these networks.
>
> We report the obtained accuracy with respect to the rank (6a) and the number of units (6b) in the paper.
>
> > Finally I think it makes sense to try some sequence datasets for which RNNs are typically used.
>
> We agree that this experiment would be a good check, however since the focus of the current work is theoretical analysis, we decided to postpone it to the future work.
>
> We also would like to note that some results of the architectures similar to the proposed on sequential datasets can be found in [Wu et al., 2016, Fig. 2] and an ICLR 2018 submission https://openreview.net/forum?id=HJJ0w--0W
>
>  > * In p7 it would help readers to point out that B^{(s,t)} is an algebraic subset because it is an intersection of M_r and the set of matrices of rank at most q^{d/2} - 1, which is known to be algebraic.
> Thank you for this remark, we have added this point to the proof.

---

### Author Response · Authors · 2018-01-02
**Update**

We would like to thank the reviewers for their time and effort to make our work better. To address the raised concerns we answered each reviewer in individual messages below and updated the paper in the following ways:

1) We have added a less formal explanation of the example constructed in the proof of Theorem 1.
2) We have added values of the hyperparameters used for the numerical experiments.
3) We have added a discussion on generalizing Theorem 1 to the case of shared TT-cores.

---

### Decision · Program_Chairs · 2018-01-29
**ICLR 2018 Conference Acceptance Decision**

**Decision:**

Accept (Poster)

**Comment:**

This paper offers a theoretical and empirical analysis of the expressivity of RNNs, in particular in comparison to TT decomposition. The reviewers argued the results was interesting and important, although there were issues with clarity of some of the explanations. More critical reviewers argued the comparison basis with CP networks was not "fair" in that their shallowness restricted their expressivity w.r.t. TT. The experiments could be strengthened by making the explanations surrounding the set up clearer. This paper is borderline acceptable, and would have benefited from a more active discussion between the reviewers and the author. From reading the reviews and the author responses, I am leaning towards recommending acceptance to the main conference rather than the workshop track, as it is important to have theoretical work of this nature discussed at ICLR.